# Characteristics of a Miniature Fiber-Optic Inorganic Scintillator Detector for Electron-Beam Therapy Dosimetry

**DOI:** 10.3390/s25144243

**Published:** 2025-07-08

**Authors:** Zhuang Qin, Ziyin Chen, Bo He, Weimin Sun, Yachen Gao

**Affiliations:** 1School of Electronic Engineering, Heilongjiang University, Harbin 150080, China; 2020013@hlju.edu.cn; 2Comprehensive Cancer Center, First Affiliated Hospital of Harbin Medical University, Harbin 150001, China; chenziyin1020@126.com; 3Key Lab of In-Fiber Integrated Optics, Ministry Education of China, Harbin Engineering University, Harbin 150001, China; _hebo@hrbeu.edu.cn (B.H.); sunweimin@hrbeu.edu.cn (W.S.)

**Keywords:** electron-beam, inorganic scintillator, fiber sensor

## Abstract

**Highlights:**

**What are the main findings?**
This study showed that the miniature fiber-optic inorganic scintillator detector demonstrates excellent dose rate linearity and repeatability when measuring electron beam doses in radiotherapy, and also performs well in measuring beam profile curves.This miniature detector shows significant discrepancies when measuring PDD curves compared to the true PDD curves; however, the results can be corrected through calibration curves.

**What is the implication of the main finding?**
Completing this work endows the miniature detector with the potential for application in an electron-beam radiotherapy environment, thereby contributing to the treatment of superficial cancers such as skin cancer.

**Abstract:**

Over the past few decades, electron beams have been widely used to treat malignant and benign tumors located in the superficial regions of patients. This study utilized an inorganic scintillator (Gd_2_O_2_S:Tb)-based radiation detector to test its response characteristics in an electron-beam radiotherapy environment, in order to determine the application potential of this detector in electron-beam therapy. Owing to the extremely high time resolution of this inorganic scintillator detector (ISD), it is even capable of measuring the pulse information of electron beams generated by the accelerator. The results indicate that for certain accelerator models, such as the IX3937, the pulse pattern of the output electron beam is notably different from that during the output of X-rays, showing no significant periodicity. The experimental results also demonstrate that this ISD exhibits excellent repeatability and dose linearity (R^2^ of 0.9993) when measuring electron beams. Finally, the PDD (Percentage Depth Dose) curves and OAR (Off-Axis Ratio) curves of the ISD were also tested under electron-beam conditions at 6 MeV and 9 MeV, respectively.

## 1. Introduction

Over the past few decades, electron beams have been utilized for the treatment of both malignant and benign tumors located in superficial regions of the patient, as well as for intraoperative radiotherapy [1,2,3,4,5]. The depth-dose of the electron beams rapidly falls off with depth resulting in a very small dose delivered to the healthy tissue downstream of the maximum range of the electron beam [6,7].

Defined as the amount of energy absorbed per unit mass of the irradiated material, the absorbed dose can induce a variety of effects, depending on its magnitude. For accurate dose calculations, treatment planning systems as well as monitor unit calculation algorithms require accurately measured beam data. However, unlike photon beam dosimetry, the ionization chamber readings of the electron beams are not linearly correlated with the dose because of the change in the water-to-air stopping power ratio with depth. Task Group 25 (TG-25) outlined recommended measurement techniques for obtaining the fundamental information necessary for the acceptance testing and treatment planning of new accelerators utilizing therapeutic electron beams [8]. Additionally, TG-25 provided comprehensive guidance on various aspects of clinical electron-beam dosimetry, including but not limited to thermoluminescent dosimetry (TLD), diode dosimetry, film dosimetry, electron source positioning, field shaping and shielding, percentage depth-dose measurements, and assessments of beam flatness and symmetry. In recent years, the primary focus of standards laboratories worldwide has transitioned from establishing standards for exposure or air kerma to developing standards for absorbed dose to water [9]. This shift reflects a growing emphasis on more accurate and clinically relevant dosimetric measurements in radiation therapy applications.

A plastic optical fiber detector based on inorganic scintillating material terbium-doped gadolinium oxysulphide (Gd_2_O_2_S:Tb, terbium-doped gadolinium oxysulfide) was developed by Sun et al. [10]. The inorganic scintillator detector (ISD) features an extremely small sensitive volume, exhibits excellent repeatability, and demonstrates perfect dose linearity (R^2^ value of 0.9999) when measuring X-rays. These characteristics indicate its potential as an in vivo detector for applications such as brachytherapy or intraoperative radiotherapy.

In this study, we conducted extensive tests using the ISD under electron-beam conditions, including measurements of electron-beam response characteristics, dose linearity, percentage depth-dose (PDD) curves, and off-axis ratio profiles, to evaluate the application potential of this inorganic scintillator material as a detector in electron-beam radiotherapy.

## 2. ISD and Fabrication

The structure of the ISD is shown schematically in Figure 1. The ISD was constructed using an SH2001-J (ESKA, Mitsubishi Rayon Co., Chiyoda-ku, Tokyo, Japan) polymethyl methacrylate (PMMA) plastic optical fiber with a core and cladding overall diameter of 500 µm. The core was micro-machined to create a 300 µm diameter and 1 mm deep hole at the tail end of the fiber [10]. This structure ensures sufficient signal intensity while minimizing the sensitive volume as much as possible. Gd_2_O_2_S:Tb was filled and packaged in the small hole using an epoxy resin adhesive. The principle of the ISD described in this paper relies on the conversion of the incident radiation (X-ray or electron beam) dose to a measurable optical signal in the visible wavelength range (490 nm, 545 nm and 590 nm), as shown in Figure 2, and the principle has been described by McCarthy (2014) [11].

The equipment used in this investigation is shown schematically in Figure 3a which details a plastic fiber-optic dosimeter submerged in a PTW MP3-M (Physikalisch-Technische Werkstätten [PTW], Freiburg, Germany) motorized water phantom in a radiotherapy bunker room. The PTW MP3-M can automatically complete the dosimeter positioning and movement, which can greatly reduce the measurement time. The software of PTW MP3-M (MEPHYSTO mcc 3.3) was used to create the scanning queue and to process the data after acquisition. The signal generated by the inorganic scintillator propagates to another tip of the optical fiber which is in the control room. The distal end of the fiber was carefully polished and machined and then connected to an MPPC C11208-350 Avalanche Photodetector Array (Hamamatsu, Hamamatsu City, Shizuoka Pref., Japan) where the intensity of the visible fluorescent light signal was measured. The advantage of utilizing the MPPC device employed in this investigation lies in its ability to operate with a very short gate time, which is 0.1 ms, which was sufficient for detecting individual pulses of the linac beam. All measurements were carried out with an electron beam delivered by a Varian IX3937 Linear Accelerator (linac, Palo Alto, CA, USA) at the External Radiation Beam Therapy clinic of the First Affiliated Hospital of Harbin Medical University, Harbin. Figure 3b shows a working photograph of a dosimeter measured in a tank.

## 3. Results and Discussion

### 3.1. Pulse Response Characteristics of Electron Beam

The ISD was positioned in the beam path at a source-to-surface distance of 100 cm, which represents a standard setup for oncology QA procedures. The ISD was then irradiated by the linac beam from directly above (0° Gantry Rotation) with the operating conditions of a 6 MeV electron beam and a 500 monitor units/min (MU/min) dose rate, for a duration of 24 s.

Figure 4a shows the optical intensity (photon counts) for consecutive macro-pulses of beam radiation, over a duration of 24 s. The excellent time resolution of the MPPC detector (0.1 ms gate time) has meant that the data captured using the device allows the individual electron-beam micro-pulses of the linac to be detected. By zooming in on 20–20.01 s of data, we can see that a single micro-pulse is made up of multiple data points in Figure 4b. Each point is the number of fluorescent photons received by the MPPC in 0.1 ms. The individual pulses here do not fully represent the shape of electron pulses; the falling edge observed is actually determined by the decay time of fluorescence.

In order to illustrate the average effect of the pulses, the red trace in Figure 4 corresponds to the 1000-point moving average imposed on the data (the blue trace in Figure 4). By comparing the blue trace and the red trace in Figure 4, it can be observed that the blue trace shows an increasing trend in the first three seconds, while the red trace remains relatively stable. This appears to present a contradiction. To explain this phenomenon, we extracted data from the intervals of 1–2 s and 20–21 s for further analysis, as shown in Figure 5. The 1 s integration period was chosen to ensure that a sufficient number of micro-pulses would be captured to observe the pattern of the electron-beam pulses. It is evident that although the micro-pulse intensity in Figure 5a is relatively low, the frequency of micro-pulses is significantly higher than that in Figure 5b. Within a one-second timeframe, there are 63 micro-pulses in Figure 5a, whereas only 45 micro-pulses are observed in Figure 5b. This ensures that the red trace representing the average remains stable throughout the process, both during the initial three seconds of micro-pulse instability and the subsequent 20 s when the micro-pulse intensity is relatively constant. This phenomenon, referred to as DRS (dose rate servo), involves a type used in some Varian machines called the pulse drop servo [12], which regulates the dose rate by dropping individual micro-pulses.

The data presented in Figure 4a were utilized to calculate the accumulated dose as recorded by the sensor. This calculation was performed by integrating the averaged dose values on a point-by-point basis. The results of Figure 6 show the linear increase in dose measured by the ISD during the application of the irradiation in macro-pulses (as shown in Figure 4a) from the linac, with a rate of increase of 2.94 × 10^5^ per second, and the final value of the accumulated dose reaches 6.73 × 10^6^.

From Figure 5, it can also be observed that the micro-pulses of the electron beam do not exhibit periodicity. This phenomenon is markedly different from the pattern observed when the linac emits X-rays, where the frequency of the pulses remains stable. We conducted measurements under the same experimental conditions except for changing the radiation to X-rays with a dose rate of 600 MU/min, and the results are shown in Figure 7. This indicates that the repetition rate (frequency) of the X-ray micro-pulses is 360 Hz, which was reported by O’Keeffe (2016) [13].

The pattern of micro-pulse delivery over a time period of 2 s was selected and recorded at a selected time of approximately 10 s, which ensures that the recording period fully coincides with the active beam delivery phase of the macro-pulse. This was repeated and recorded in the same manner as Figure 5, but with the dose rate varied from 100 MU/min (as in Figure 5) to 500 MU/min in steps of 100 MU/min. The data zoomed in to the 2 s window at the time of 10 s, corresponding to every case, are shown, respectively, in Figure 8a–e. As can be seen from Figure 8, the micro-pulses at any dose rate, over the full measurement range of 100 to 500 MU/min, do not exhibit periodicity. This phenomenon is also entirely different from the regularity of X-ray emission micro-pulses [13].

If the number of complete micro-pulses captured in each frame from Figure 8a–e is counted and the accumulated value within two seconds is calculated, the following table can be constructed (see Table 1). Although the micro-pulses of the electron beam do not exhibit periodicity, the data contained in Table 1 clearly demonstrate that there is a linear relationship between the number of complete micro-pulses captured in a 2 s window and the dose rate, as shown in Figure 9a. As the dose rate increases, a noticeable increase in the number of individual micro-pulses can be observed within a 2 s time window. Similarly, there is a clear linear relationship between the accumulated intensity within two seconds and the dose rate, as shown in Figure 9b. However, both the linearity of the pulse count and the linearity of the accumulated intensity are slightly lower (R^2^ = 0.9941 and 0.9966). This is mainly due to the instability of the pulses, as shown in Figure 4a. If we compare the total accumulated count within 24 s with the dose rate, we find that the linearity of the dose rate is very excellent, as shown in Figure 10. A linear regression analysis shows that the R^2^ value in this case is 0.9993. Also, the intercept is very close to the (0, 0) point, the y axis intercept being at (0, −43548). This corresponds to a value of 0.65% of the maximum value on the y axis.

The results in Figure 10 unequivocally demonstrate that although the amplitude of individual micro-pulses of electron-beam irradiation delivered by the linear accelerator may fluctuate, if the amplitude data are averaged over a sufficiently long interval (in this case 2 s), the fluctuations become insignificant, and the average amplitude value can highly accurately reflect the dose rate. It is crucial to confirm the mode of radiation delivery in pulse form, as it is essential for ensuring the accuracy of dose measurement by the ISD, especially if the ISD is intended for use as a real-time dosimeter in clinical applications.

### 3.2. Dosimeter Repeatability

As a potential one-time, replaceable real-time dosimeter, it is necessary to evaluate whether the ISD has repeatability within a short period of time. The ISD was tested to determine its reproducibility over five exposures with each exposure at 200 MU at a dose rate of 400 MU/min for 30 s with the operating condition of a 6 MeV electron beam. The data were recorded using an MPPC with a gate time of 100 ms. Figure 11 illustrates the exceptional repeatability of the measurements across five exposures. During each phase output period, the monitored peak intensity displayed minor fluctuations, and the average intensity consistently remained around 120,000. The area under each curve during the beam’s active phase corresponds to the dose received during exposure. The integrated intensity obtained in this study is presented in Figure 12, and the data for the five-exposure test are summarized in Table 2. Given that the dose for each exposure was identical, the integrated intensity for each exposure should also be consistent. As shown in Table 2, the maximum error percentage was 1.47%, which confirms the excellent repeatability of the dosimeter in monitoring radiation doses.

### 3.3. Depth-Dose Experiment

The ISD’s depth-dose measurement was performed with a dose rate of 400 MU/min at 6 MeV in depth from 0 cm to 5 cm in steps of 1 mm. The results from these measurements were compared with similar measurements for the PTW31010 0.125cc cylindrical Ionization Chamber (Physikalisch-Technische Werkstätten [PTW], Freiburg, Germany). This PTW 31010 miniature ionization chamber was calibrated at the National Institute of Metrology, China, in December 2023. It has been used for routine absolute dose measurements in G3 CyberKnife treatments and has demonstrated good repeatability. Figure 13 shows the results of the depth-dose profile for the 6 MeV electron beam measured using ISD and IC. The Dmax for the 6 MeV beam measured using the IC occurred at a depth of 15 mm, while the ISD’s D_max_ was found to be at around 7 mm. When using the ISD to measure the point of maximum dose for an electron beam, a significant forward shift is observed compared to the point of maximum dose measured by the IC. This is contrary to the results obtained when measuring X-rays, as shown in Figure 14.

The phenomenon of the maximum dose point shifting backward when ISD is used to measure X-ray has been reported and explained by Qin et al. (2019) [14] and He et al. (2024) [15]. This is because the absorption of radiation by water or human tissues is caused by the energy loss of secondary electrons, and there is only a small number of backscattered electrons in the shallow layer of water. But for ISD, it can directly absorb the scattered photons to emit fluorescence, resulting in a larger signal output. The difference in response mechanisms between water and inorganic scintillators ultimately leads to the phenomenon of the maximum dose point shifting backward under X-ray irradiation.

For the electron beam, according to the formula for collisional stopping power of electrons:(1)$${\left(\frac{S}{\rho }\right)}_{col}=2\pi r_{e}^{2}N_{e}\frac{\mu_{e}}{\beta^{2}}\left[\ln\frac{E^{2}\left(E+2\mu_{e}\right)}{2\mu_{e}I^{2}}+\frac{E^{2}/8-\left(2E+\mu_{e}\right)\mu_{e}\ln2}{{\left(E+\mu_{e}\right)}^{2}}+1-\beta^{2}-\delta \right]$$

Ionization loss increases slowly with electron energy at high energies. Conversely, at low energies, ionization loss is inversely proportional to electron energy. When high-energy electrons propagate in water, their range is very short, and energy is rapidly lost, converting them into low-energy electrons. As ionization loss increases significantly, the response of IC quickly rises to reach the maximum dose point. With the depletion of electron energy, ionization loss decreases rapidly, causing the PDD curve to drop sharply and eventually approach zero.

For ISD, based on the formula for the radiative stopping power of electrons:(2)$${\left(\frac{S}{\rho }\right)}_{rad}\propto \frac{z^{2}Z^{2}}{m^{2}}NE$$

When electrons pass near atomic nuclei, they lose energy, which is emitted as bremsstrahlung radiation under the influence of the Coulomb field. This radiative loss is proportional to the electron energy and the material’s Z^2^. At shallow water depths, the combination of high-energy electrons and the high atomic number (Z) of the scintillator leads to a rapid generation of substantial bremsstrahlung within the ISD, exceeding that produced in water (over 4–20 MeV energy range). Owing to the luminescence mechanism of Gd_2_O_2_S:Tb, these photons can be directly absorbed to generate fluorescence, resulting in a steeper and faster rise of the PDD curve for the electron beam compared to the actual curve measured by an ionization chamber [14,15]. As the electron energy decreases rapidly, the number of newly generated bremsstrahlung photons diminishes quickly, leading to a decline in the PDD curve and causing the maximum dose point to shift forward.

At water depths greater than 30 mm, the response intensity measured by the ISD remains higher than expected and does not decrease to zero. This is due to the fact that although the intensity of bremsstrahlung radiation is low, the propagation distance of bremsstrahlung photons is long, allowing them to penetrate depths much greater than 30 mm. For the ISD, due to the dependence of the photoelectric effect on the atomic number, low-energy bremsstrahlung photons can still interact with Gd_2_O_2_S:Tb via the photoelectric effect, producing a certain number of secondary electrons and resulting in some dose absorption. Therefore, there is always a certain level of intensity at depths below 30 mm. In addition, the generation of Cerenkov may also cause some response. High-energy electrons in water can produce Cerenkov radiation, which, although relatively weak in intensity, still contributes to a measurable level of intensity at water depths below 30 mm. The actual impact of Cerenkov radiation will be carefully evaluated in our future work using bare optical fibers and Monte Carlo simulations.

As the ratio between the radiation dose received by water (D_IC_) and the dose absorbed by the scintillator (D_ISD_) confirms, the depth-dose response measured by the ISD can be calibrated by multiplying the correction coefficient. Figure 15 shows the curve of this ratio as a function of depth.

The energy of the electron beam was adjusted to 9 MeV, and the PDD experiment was repeated under identical experimental conditions, as shown in Figure 16. It can be observed that as the electron energy increases, the range of the electrons becomes longer, resulting in a significant backward shift in the maximum dose point in the PDD curves measured by both ISD and IC.

### 3.4. Beam Profiling Experiment

The off-axis ratio curves of the ISD in the x-axis direction were experimentally tested under a 10 × 10 cm^2^ radiation field with 6 MeV and 9 MeV electron beams. These measurement data were compared with those obtained using a 0.125cc cylindrical Ionization Chamber, as shown in Figure 17.

From Figure 17a, it can be observed that under the irradiation condition of 6 MeV electron beams, the in-of-field region measurement results of ISD are highly consistent with those of IC, and both dose responses tend to flatten. In the penumbra region, the curve measured by ISD is also consistent with that of IC. However, there is a certain difference in the out-of-field region, which is similar to the reason for the non-zero response intensity observed in PDD experiments at depths greater than 30 mm. This is mainly due to bremsstrahlung radiation generated by the interaction of high-energy electrons with water, which propagates into the out-of-field region and interacts with the scintillation material, thereby producing a certain intensity.

As shown in Figure 17b, we can see that under the irradiation condition of 9 MeV electron beams, the test results of ISD in both the in-of-field and out-of-field regions are similar to those under the 6 MeV condition. However, there is a noticeable difference in the penumbra region, where the curve measured by ISD shows a slower rate of decline compared to the results obtained by IC. The reason for this difference is evident: as the electron energy increases, the probability of bremsstrahlung radiation production significantly increases, as shown in Equation (2). This additional X-ray contamination interacts with the scintillation material even in regions where electrons cannot easily reach, leading to an over-response phenomenon and increasing the width of the penumbra region.

From the OAR experiment of this electron beam, it can be seen that under the 6 MeV condition, ISD demonstrates a considerable level of accuracy in measuring radiation dose; however, under the 9 MeV electron-beam condition, a certain degree of calibration is required for ISD to function properly.

## 4. Conclusions

In this study, we investigated the response characteristics of an inorganic scintillating fiber dosimeter under electron-beam radiation conditions. Owing to the excellent time resolution of the MPPC, we used ISD to test the pulse response of electron beams and discovered that the IX3937-type accelerator exhibits different output modes when delivering electron beams versus X-rays. When outputting X-rays, the accelerator demonstrated distinct periodic output characteristics, which were not observed during electron-beam output. In this way, the accelerator can only adjust the dose rate stability through DRS, which has also been verified by the experimental results of ISD.

The reproducibility, PDD, and OAR profile response of the ISD were also tested. The results of these experiments showed that the ISD exhibited good repeatability, with a maximum percentage error of 1.47% and very good dose linearity (R^2^ of 0.9993). In the PDD experiment, bremsstrahlung radiation is generated when electron-beam energy exceeds 4 MeV due to interaction with water; this X-ray contamination can interact directly with Gd_2_O_2_S:Tb to produce visible light, thereby significantly altering the shape of the PDD curve. As a result, we had to calculate its correction factor curve to enable the ISD to accurately measure the PDD curve. In the OAR experiment, we found that under 6 MeV electron-beam irradiation, the results measured by ISD and IC were highly similar both in the field and in the penumbra region, with only slight differences observed out-of-field due to the effect of bremsstrahlung radiation. However, under 9 MeV electron-beam irradiation, ISD and IC showed consistent results only in the in-field region, both exhibiting a flat dose distribution characteristic. In the penumbra region, the increased X-ray contamination significantly broadened the penumbra width, causing noticeable differences between the OAR curves measured by ISD and those obtained by IC. Moreover, Cerenkov radiation may also affect the PDD and OAR curves. The investigation of the impact of these physical processes on ISD measurement results under different experimental conditions will be one of the main focuses of our future work.

The experimental results indicate that although the ISD exhibits good dose linearity and reproducibility in an electron-beam therapy environment, significant differences still exist compared to ICs in various dose distribution measurements. This necessitates extensive determination of various correction factors for the ISD before it can be applied to actual electron-beam dosimetry. Completing this work endows the ISD with the potential for application in an electron-beam radiotherapy environment, thereby contributing to the treatment of superficial cancers such as skin cancer.

## Figures and Tables

**Figure 1 sensors-25-04243-f001:**
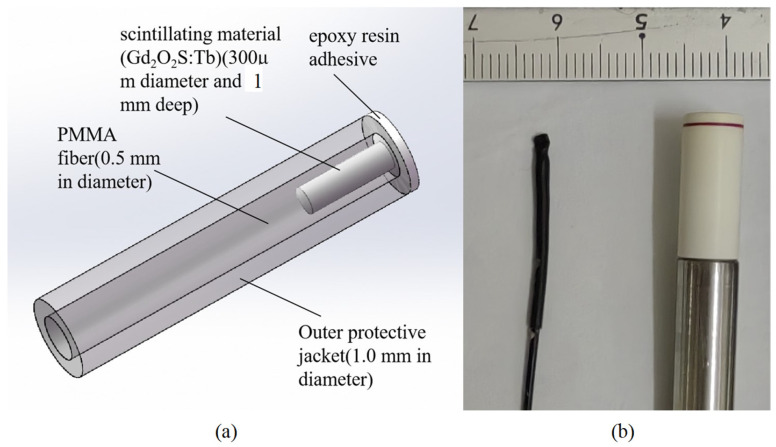
(**a**) Schematic representation of the inorganic scintillator detector for external beam radiotherapy. (**b**) Inorganic scintillator detector head dimension (left) compared with miniature ionization chamber PTW31010 (Physikalisch-Technische Werkstätten [PTW], Freiburg, Germany) (right).

**Figure 2 sensors-25-04243-f002:**
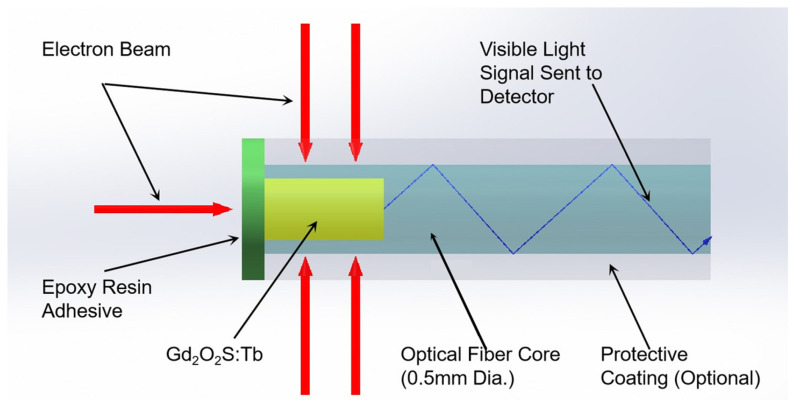
The working mechanism for the plastic optical fiber sensor.

**Figure 3 sensors-25-04243-f003:**
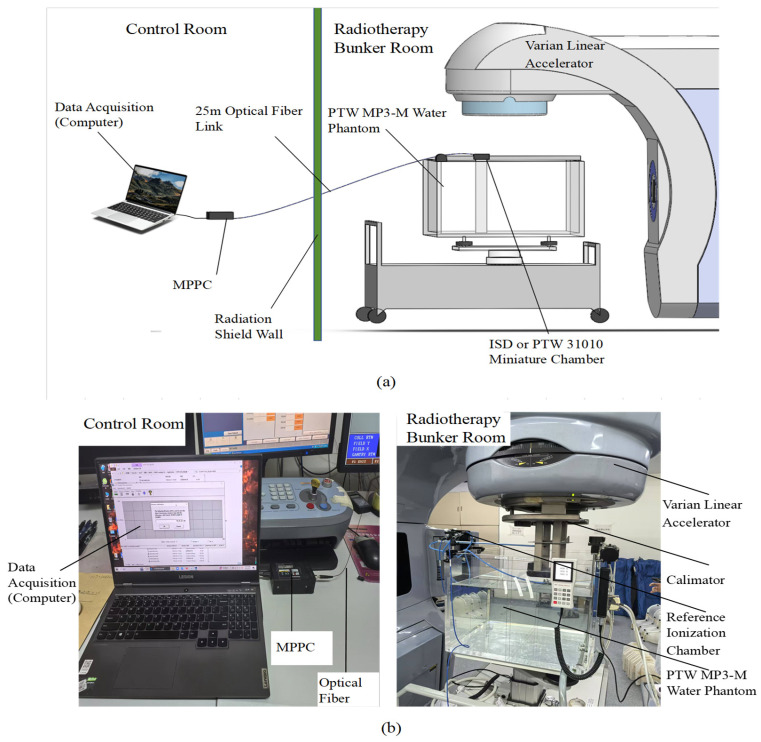
Experimental Setup: (**a**) Schematic layout. (**b**) PTW MP-M motorized three-dimensional (3D) water phantom.

**Figure 4 sensors-25-04243-f004:**
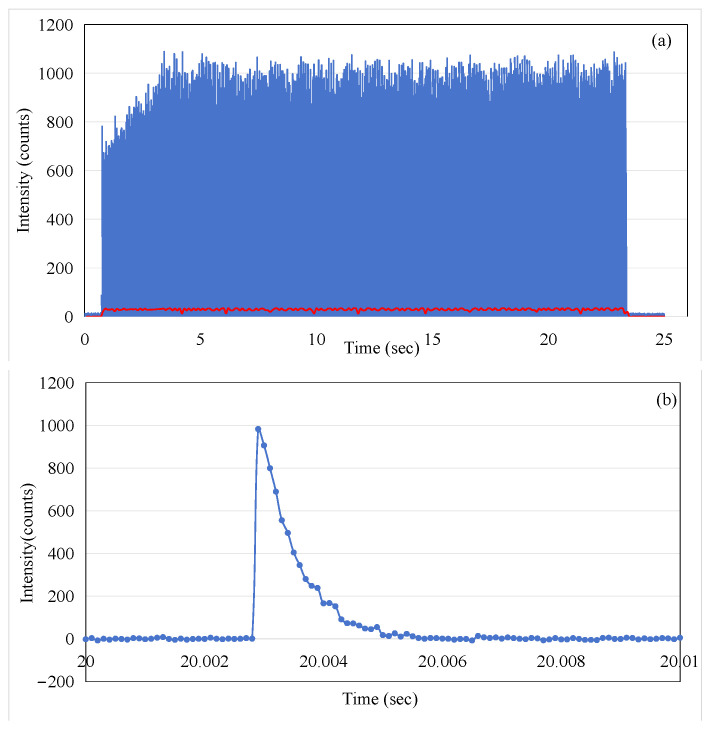
(**a**) The ISD output intensity captured over the 24 s of beam radiation for consecutive electron-beam pulses and the result of a 1000-point moving average applied (shown as the red line trace); (**b**) the output micro-pulses of the Linac at the time t = 20 s in (**a**).

**Figure 5 sensors-25-04243-f005:**
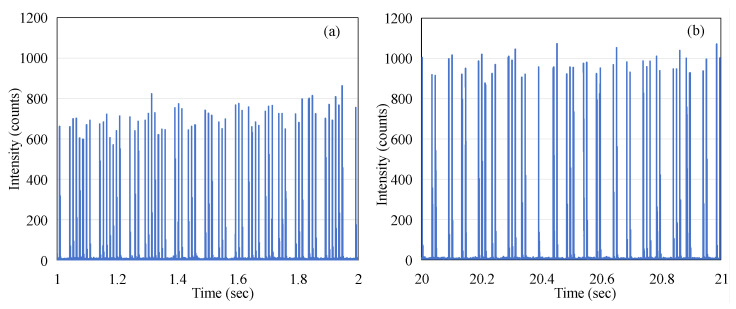
(**a**) The output micro-pulses of the electron beam at the time t = 1 s in Figure 4a; (**b**) the output micro-pulses of the electron beam at the time t = 20 s in Figure 4a.

**Figure 6 sensors-25-04243-f006:**
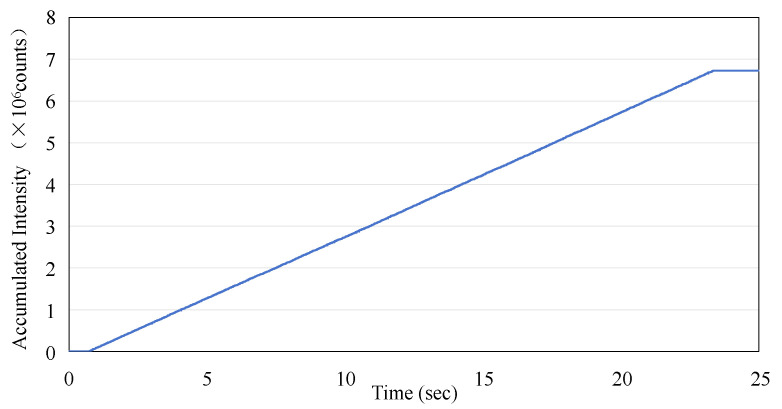
The accumulated intensity measurement using the ISD versus time as derived from Figure 4a.

**Figure 7 sensors-25-04243-f007:**
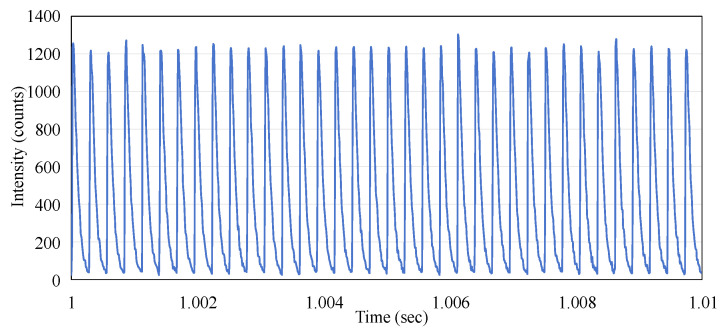
The time-resolved output pulses of the Linac X-ray beam.

**Figure 8 sensors-25-04243-f008:**
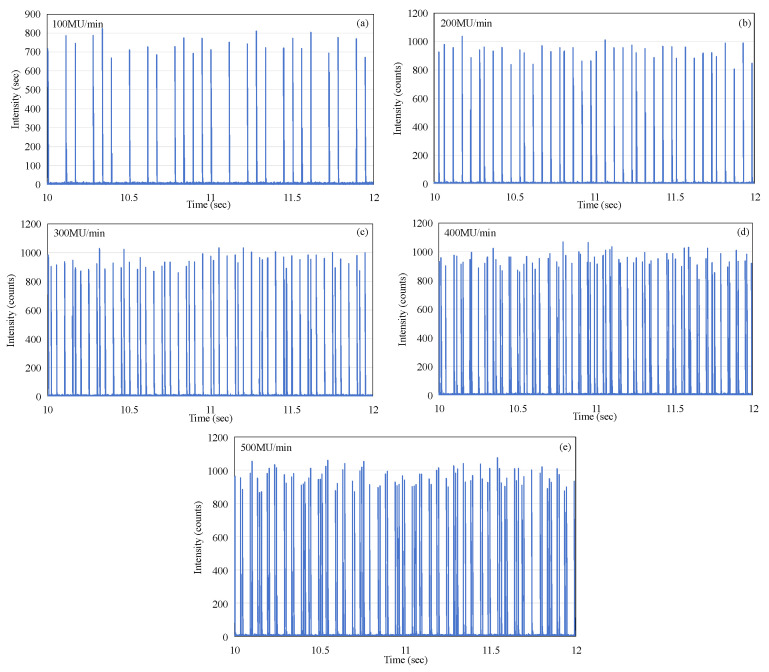
The output micro-pulses of the electron beam at the time t = 10 s (**a**) 100 MU/min, (**b**) 200 MU/min, (**c**) 300 MU/min, (**d**) 400 MU/min, and (**e**) 500 MU/min.

**Figure 9 sensors-25-04243-f009:**
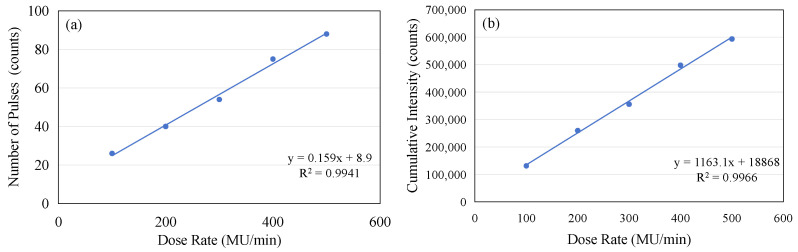
(**a**) The linear relationship between the number of micro-pulses captured in a 2 s window and the dose rate. (**b**) The linear relationship between the accumulated intensity in a 2 s window and the dose rate.

**Figure 10 sensors-25-04243-f010:**
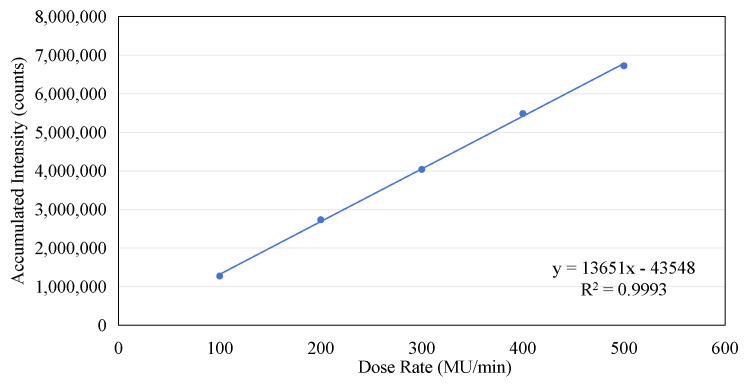
The linear relationship between the accumulated intensity and the dose rate.

**Figure 11 sensors-25-04243-f011:**
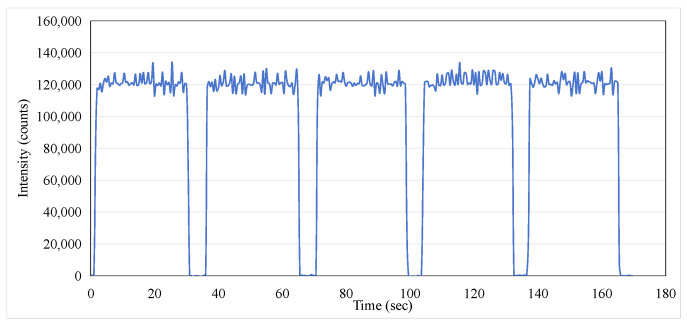
Optical intensity for repeated exposures of 200 MU at a dose rate of 400 MU/min at 6 MeV electron beam.

**Figure 12 sensors-25-04243-f012:**
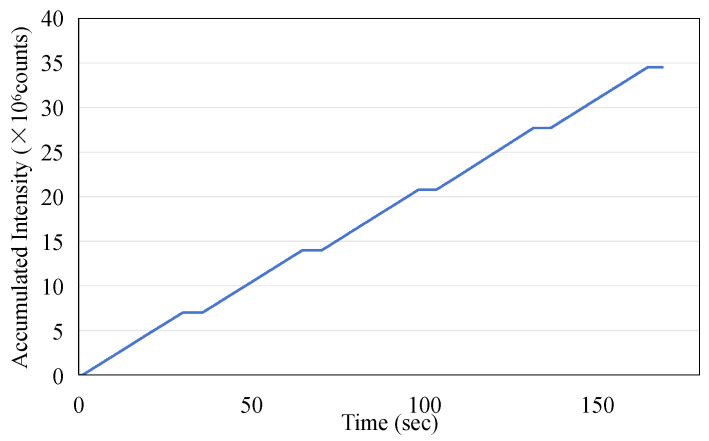
The accumulated dose measurement using the ISD derived from Figure 11.

**Figure 13 sensors-25-04243-f013:**
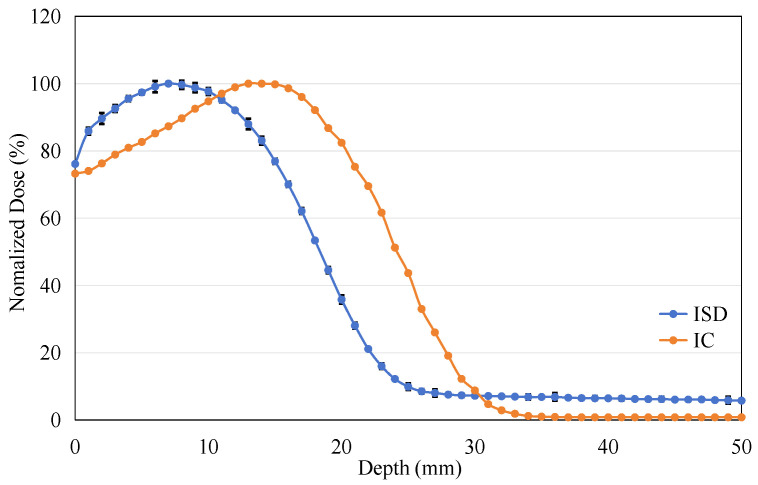
The percentage depth dose: measured on ISD and measured on Ionization Chamber for 6 MV Electron-beam.

**Figure 14 sensors-25-04243-f014:**
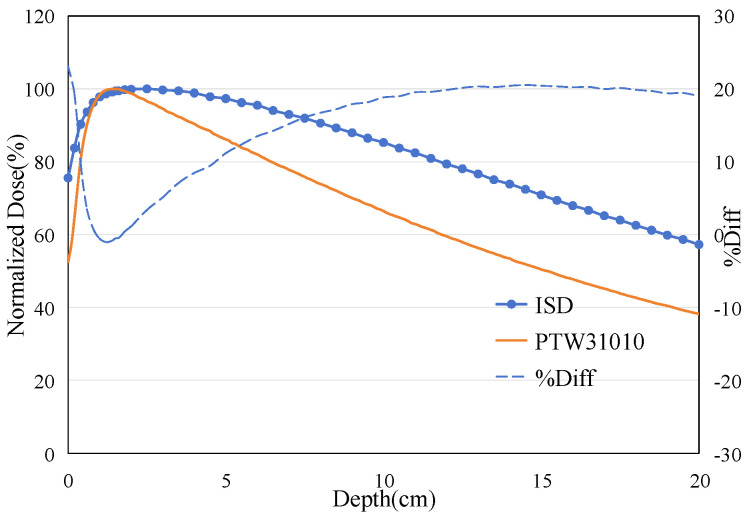
The percentage depth dose: measured on ISD and measured on Ionization Chamber for 6 MV X-ray beam.

**Figure 15 sensors-25-04243-f015:**
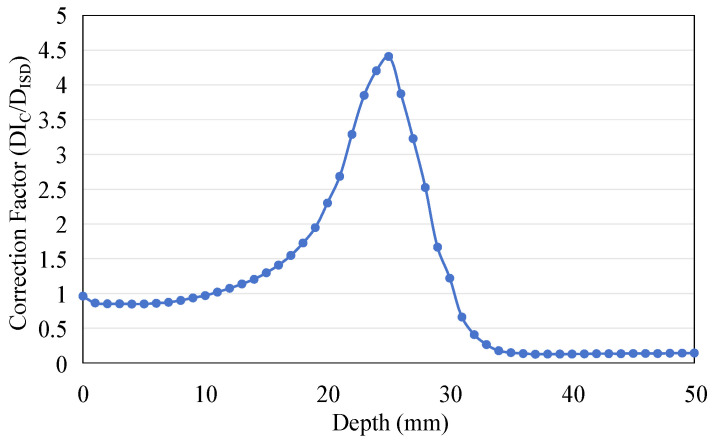
The ratio between the dose deposited in water (D_IC_) and the dose deposited in the scintillator (D_ISD_) changes with depth.

**Figure 16 sensors-25-04243-f016:**
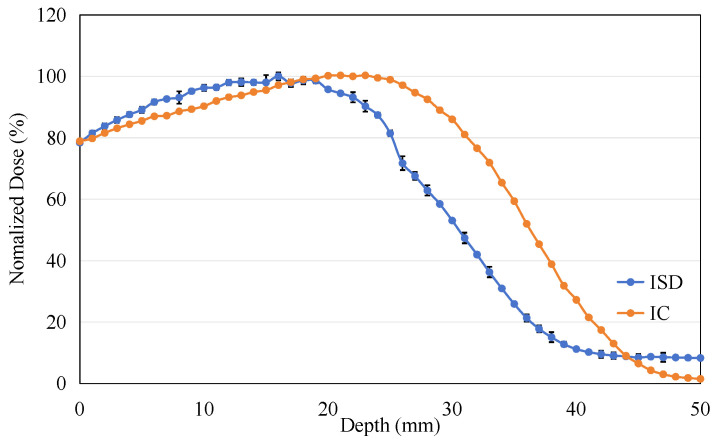
The percentage depth dose: measured on ISD and measured on Ionization Chamber for 9 MeV electron beam.

**Figure 17 sensors-25-04243-f017:**
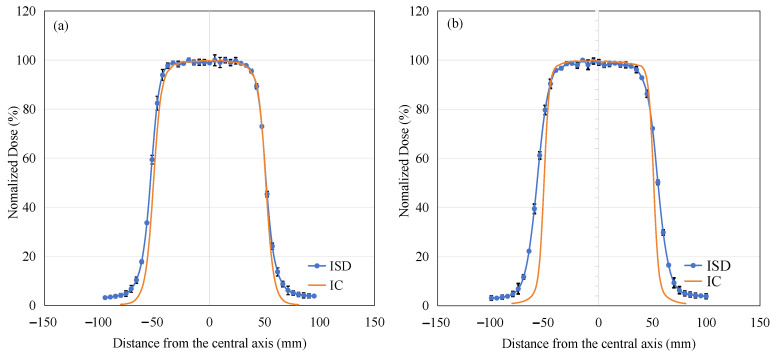
Dose profiles obtained using ISD (blue) compared to IC (orange) with (**a**) 6 MeV and (**b**) 9 MeV electron beams.

**Table 1 sensors-25-04243-t001:** Comparison of Pulse Count and Dose Rate.

Dose Rate (MU/min)	100	200	300	400	500
Number of Pulses	26	40	54	75	88
Accumulated Value (×10^5^ counts)	1.31	2.60	3.56	4.98	5.93

**Table 2 sensors-25-04243-t002:** Integrated optical intensity, and percentage error, for repeated exposures at 200 MU.

Exp. No.	Integrated Intensity	Percentage Error (%)
1	7,001,290.94	1.37%
2	6,982,753.51	1.10%
3	6,805,259.33	−1.47%
4	6,911,834.98	0.08%
5	6,831,592.33	−1.09%

## Data Availability

The datasets generated during and analyzed during the current study are available from the corresponding author upon reasonable request.

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
