# Peer review of "Characteristics of a Miniature Fiber-Optic Inorganic Scintillator Detector for Electron-Beam Therapy Dosimetry"

_sensors, 2025, doi:10.3390/s25144243_

Round 1

Reviewer 1 Report

Comments and Suggestions for Authors

The manuscript. Characteristics of a Miniature Fiber-Optic Inorganic Scintillator Detector for Electron-Beam Therapy Dosimetry, presents an experimental evaluation of an ISD detector applied to electron beam dosimetry. Showing good linearity and repeatability. However, some methodological limitations must be addressed before publication. The points are listed below.

1 Could the authors specify how compensation for these specific physical effects will be incorporated into the development and application of ISD correction factors and calibration curves?

2 How do the authors plan to address a lack of Cerenkov radiation quantification, the absence of uncertainty analysis, and the lack of ionization chamber validation to strengthen the clinical applicability of the ISD?
The study mentions Cerenkov radiation without correction, uses only percentage errors and R² without error bars or systematic uncertainties, and does not detail the calibration of the PTW31010 or its inter-day repeatability.

3 Could the authors detail how the 1000-point moving average specifically affects the inherent temporal resolution of the detector and whether this filtering technique introduces any artifacts into the representation of the dynamics of individual micropulses or macropulses in the analysis?

I recommend that the manuscript be considered for publication once the authors adequately address the points outlined above. Although the work is interesting and has potential clinical applicability, certain aspects need to be clarified and justified before final acceptance, so I recommend minor revisions.

Reviewer 2 Report

Comments and Suggestions for Authors

The paper presented looks interesting and useful for specialists dealing with fiber medical sensors to control radiation level during therapy treatment. The developed sensor model was tested from many points of view importat from practice- pulse regime, comparision electron beam and X-ray radiation, accuracy and linearity of sensor. The materials are presented clear and detailed, the only my worry is that signatures on the drawing are unreadeble ? at least at my PC. In general the paper could be published in a present form.

Reviewer 3 Report

Comments and Suggestions for Authors

Electron beams have been widely used for treating superficial tumors. 
And this study tested an inorganic scintillator detector (Gd2O2S) in radiotherapy to evaluate its potential. The detector's high time resolution enabled measurement of electron beam pulses, revealing non-periodic patterns in certain accelerators like the IX3937. 

Results showed excellent repeatability, dose linearity (R²=0.9993), and successful PDD/OAR curve testing at 6 MeV and 9 MeV. I recommend major revision for this submission.

  1. The working mechanism for the plastic optical fiber sensor should be explained by including one schematic;
  2. In figures 3-16, all of the texts along the vertical axis are obscured;
  3. Line 293 and figure 13, the length unit (3 cm) should be unified with other figures (30 mm).
  4. Figure 2 should be reworked to illustrate the different parts in both schematic and pictures of the experimental setup.

Round 2

Reviewer 3 Report

Comments and Suggestions for Authors

The modifications are satisfied. I recommend the acceptance for this submission in its present form.